

# Improved genetic algorithm optimized LSTM model and its application in short-term traffic flow prediction

Junxi Zhang, Shiru Qu, Zhiteng Zhang and Shaokang Cheng

School of Automation, Northwestern Polytechnical University, Xi'an, China

## ABSTRACT

Considering that the road short-term traffic flow has strong time series correlation characteristics, a new long-term and short-term memory neural network (LSTM)-based prediction model optimized by the improved genetic algorithm (IGA) is proposed to improve the prediction accuracy of road traffic flow. Firstly, an improved genetic algorithm (IGA) is proposed by dynamically adjusting the mutation rate and crossover rate of standard GA. Secondly, the parameters of the LSTM, such as the number of hidden units, training times, gradient threshold and learning rate, are optimized by the IGA. Therefore, the optimal parameters are obtained. In the analysis stage, 5-min short-term traffic flow data are used to demonstrate the superiority of the proposed method over the existing neural network algorithms. Finally, the results show that the Root Mean Square Error achieved by the proposed algorithm is lower than that achieved by the other neural network methods in both the weekday and weekend data sets. This verifies that the algorithm can adapt well to different kinds of data and achieve higher prediction accuracy.

# INTRODUCTION

Traffic congestion is a hotspot in the field of smart city and intelligent transportation (*El Hamdani, Benamar & Younis, 2020*; *Jun & Yongjun, 2021*). Intelligent transportation system (ITS) can effectively strengthen the coordination between people, vehicles and roads, and forms a safe, clean and environment-friendly comprehensive transportation system by comprehensively applying advanced information technology, computer technology, sensing technology and artificial intelligence to transportation and service control (*Kalamaras et al., 2018*). Short-term traffic flow prediction is the key technology of ITS to solve traffic congestion and guidance. By collecting the temporal and spatial characteristics of historical traffic flow data, the short-term traffic flow in the future can be accurately predicted to obtain real-time traffic status, so as to provide decision support for traffic dredging and traffic control (*Lu, Sun & Qu, 2015*; *Ghofrani et al., 2018*).

There are three common model prediction methods: parameter based prediction, shallow machine learning prediction and deep learning prediction. The typical parameter-based prediction model of ARIMA was applied to traffic flow prediction in *Kumar & Vanajakshi (2015)*. This method lacks of consideration of the temporal and spatial

Corresponding author
Junxi Zhang,
zhangjunxi@xaau.edu.cn

sequence characteristics of traffic flow. The prediction methods of shallow machine learning model including support vector machine (SVM), BP neural network, *etc.* have the defects of slow processing speed and low accuracy. In contrast, deep learning models, such as deep belief network (DBN), convolutional neural network (CNN), stacked automatic encoder neural network (SAE), cyclic neural network (RNN), long-term and short-term memory neural network (LSTM), gated cyclic unit neural network (GRU), *etc.*, have unique advantages in the processing of time series data. With the development of vehicle networking technology and artificial intelligence, deep learning model has become the hot spot of current research and been widely used in the field of traffic flow prediction. The pure deep learning model also has some inherent defects. For example, although the LSTM can fully handle to the time series characteristics of traffic flow, it is difficult to balance the depth and operation time complexity of its model. The dependence of the RNN model on long time series is difficult to deal with, and even the gradient vanishes. By combining the optimization algorithm with the deep learning algorithm, the output root mean square error can be significantly improved (*Wen, Zhang & Lu, 2019*; *Hochreiter & Schmidhuber, 1997*; *Song, Li-Jun & Man, 2012*; *Chan et al., 2012*; *Ma, Zhou & Antoniou, 2018*; *Schmidhuber, 2015*).

The parameter selection of the LSTM algorithm has a great impact on the results. The traditional methods use traversal search and control parameter adjustment, which has a large time complexity. To deal with the large amount of data and strong time dependence of road short-term traffic flow time series data. The main contributions of this article are summarized as follows:

1. We improved and optimized the genetic algorithm model (IGA), and obtained the IGA model which has higher convergence efficiency.
2. The IGA is applied to optimize the LSTM model, and the optimized algorithm is named as the IGA-LSTM. The IGA-LSTM can be used to predict the 5-min short-term traffic flow.
3. The improved genetic algorithm (IGA) is used to optimize the training times, gradient threshold, hidden layers, learning rate and other parameters of the LSTM, aiming at minimizing the output root mean square error.
4. The traffic flow data of California Highway Administration PEMS system are used to demonstrate the performance of the proposed design. Both workday and weekday data prediction results are simulated.
5. The experiment results show that the proposed IGA-LSTM model has higher prediction accuracy and faster convergence speed than the shallow machine learning algorithms like GA-BP model, PSO-BP model and pure LSTM model.

Based on the above problems, we proposed a short-term traffic flow prediction algorithm which use the improved genetic algorithm to optimize the long-short-term memory neural network. First, the crossover rate and mutation rate of the GA algorithm are adaptively adjusted and improved. Second, the framework and characteristics of the LSTM algorithm are analyzed. The improved GA (IGA) algorithm is used to optimize

number of hidden units, training times, gradient threshold and learning rate of the LSTM algorithm. Then, the optimal parameters are obtained. The IGA-LSTM short-term traffic flow prediction model is constructed. Finally, the experimental verification is carried out using traffic flow dataset form the PEMs.

## RELATED WORKS

In this section, we will review related works on deep learning method and short-term traffic flow. *Wu et al. (2018)* proposed a traffic flow prediction model based on deep neural network, which established an attention model, and obtained the correlation of historical traffic flow data through the model, so as to predict the future traffic flow. *Bogaerts et al. (2020)* proposed a CNN-LSTM neural network prediction model for short-term and long-term traffic flow, which can extract the temporal and spatial characteristics of traffic flow data at the same time. Considering the problem that the accuracy of data-driven prediction model is not high when the amount of training data is small or the noise is large. *Yuan et al. (2021)* proposed a PRGP model to strengthen the estimation of traffic flow through shadow GP, and the prediction result is better than that of simple machine learning algorithm. Considering the problem of missing data in the process of short-term traffic flow prediction, *Yang, Peng & Lin (2021)* proposed a Spatio-temporal prediction model based on original incomplete data. This method realized the interpolation of missing values, and captured time series and spatial features through long-term memory (LSTM) network and Laplace matrix (GL) model. Compared with the deep learning model of multivariable spatiotemporal traffic data set, the proposed LSTM-GL model has better robustness and prediction accuracy (*Yang, Peng & Lin, 2021*). *Sun & Kim (2021)* predicted the next position and travel time by capturing the correlation between position and time series in trajectory data through hybrid LSTM and sequential LSTM deep learning models. This work obtained higher prediction accuracy compared with hidden Markov model and LSTM model (*Sun & Kim, 2021*). *Wen, Zhang & Lu (2019)* used genetic algorithm to optimize the number of hidden layers, training times and dropout parameters of the LSTM model. Based on that, the long-term and short-term traffic flow of express way are predicted after obtaining the optimal parameter. Compared with the pure machine learning and deep learning model, the prediction mean square error obtained in *Sun & Kim (2021)* is smaller. *Hou et al. (2022)* proposed a CNN+BiLSTM model, which is the first to combine rule embedding, a parallel structure of Convolutional Neural Network (CNN) and a two-layer bi-directional long short-term memory (BiLSTM) with the selfattention mechanism. *Ali et al. (2021)* proposed a social network-based, real-time monitoring framework for traffic accident detection and condition analysis using ontology and latent Dirichlet allocation (OLDA) and bidirectional long short-term memory (Bi-LSTM). The results achieves accuracy of 97%. In the mean while, his research on traffic accident detection and sentiment analysis of transportation in *Ali, El-Sappagh & Kwak (2019)*, *Ali et al. (2019)*, *Ali et al. (2017)*, which improved the task of transportation features extraction and text classification using the bi-directional long short-term memory (Bi-LSTM) approach.

**Figure 1 Overall framework of prediction model.** The blue background represents the data processing flow, and the gray background represents the specific operation steps.

However, most of these systems use empirical values to initialize the deep learning algorithm models, which are sensitive to initial values. That is not scientifically rigorous enough. The parameters of model LSTM mostly uses traversal multi-grid search algorithm, which has high computational complexity. Also, the adjustment is complicated. In addition, traffic flow prediction is a complex system engineering, which needs to comprehensively consider spatial information and time information. Therefore, this study uses evolutionary algorithm to optimize the initial parameters of the deep learning model so as to improve the convergence efficiency and prediction accuracy of the algorithm.

# DESCRIPTION OF TRAFFIC FLOW TIME SERIES MODEL

Traffic flow can be expressed as a historical time series at a given observation location. For a certain time range, short-term traffic flow series can be predicted in the corresponding time interval. Assuming that $n$ detectors are set in a certain road section, the historical observation data of traffic flow (*vehicle/t*) of the $S$ detector in period $t$ is defined as follows:

$$X = \{Q_s(t), Q_x(t-1), Q_x(t-2), \ldots\ldots, Q_x(t-P)/s = 1, 2, 3\ldots, N\} \qquad (1)$$

where $t$ represents the current period, $P$ represents time lag, and the number of all variables is expressed as Eq. (2).

$$L = N(P+1) \qquad (2)$$

$X$ represents the traffic state of the whole road network at the current and $P$ historical times, and $Q_S(t)$ depends on time and spatial layout, which is called Spatio-temporal variable. Output $Y$ can be expressed by Eq. (3).

$$Y = Q_x(t+1) \qquad (3)$$

The prediction model is the mapping relationship between input $A$ and output $Y$, which can be expressed by Eq. (4).

$$F(t) = f(X(t) \rightarrow Y(t)) \qquad (4)$$

The overall framework of the system model is shown in Fig. 1. The traffic flow detector collects the original traffic flow data, divides and preprocesses the collected data. The missing data in the collection process is supplemented by the method of front and rear mean and normalized with the abnormal data eliminated.

The operation process of the model can be described as follows:

1. Data preparation and data preprocessing, and establishment of predictive models.
2. Optimize the crossover operator and mutation operator of the genetic algorithm to obtain the IGA algorithm.
3. The IGA algorithm is used to optimize the LSTM parameters, and the target problem is transformed into a biological evolution process. New populations are generated through operations such as crossover, mutation, and replication, and solutions with low fitness are eliminated.
4. The population is initialized and decoded, and the mean square error of the LSTM neural network is used as the fitness function. The individual of the solution is subjected to selection crossover mutation operation.
5. If the target value of the fitness function reaches the optimal value, go to the next step; otherwise, go back to step (4).
6. Obtain the fitness target value and optimal parameters. Calculate the mean square error of prediction based on the best parameters.
7. Judge the terminate conditions, if the number of iterations of the population is satisfied, stop the calculation. Now, the global optimal parameter combination of the LSTM network; otherwise, return to step (6).
8. Output the normalized data, perform error analysis, obtain the final prediction result and compare it with several other shallow layers and their learning algorithms such as the GA-BP, the PSO-BP and the LSTM, and give the final conclusion.

## THE OPTIMIZATION OF GENETIC ALGORITHM

The genetic algorithm (GA) is an adaptive global optimization probabilistic search algorithm tool. Just as described in the *Pragadeesh et al. (2019)*, *Shukla (2020)*, GA is a global search algorithm. By improving it, it can avoid local optimum.

We have analyzed the crossover rate and the mutation rate of standard GA in *Zhang & Qu (2021)*. And we have obtained the result that a too large crossover rate or a too small crossover rate will affect the search efficiency of the GA. An optimized GA has been proposed in *Jiao et al. (2019)*, in which the mutation rate and crossover rate can be adjusted appropriately. And also we introduced the variable crossover process and mutation rate parameters in *Zhang & Qu (2021)*, The adaptive crossover rate $P_C$ based on the IGA can be expressed as Eq. (5), as follows:

$$P_C = \begin{cases} \dfrac{P_{C1}(F_{max} - F')}{F_{max} - F_{min}}, F' \geq F_{avg} \\ P_{C2}, F' < F_{avg} \end{cases} \tag{5}$$

The adaptive mutation rate $P_V$ can be calculated as Eq. (6):

$$P_V = \begin{cases} \dfrac{P_{V1}(F_{max} - F)}{F_{max} - F_{min}}, F \geq F_{avg} \\ P_{V2}, F < F_{avg} \end{cases} \tag{6}$$

In Eqs. (5) and (6), $F'$ represents the maximum fitness value of the parent individual in the process of the crossover operation; $F$ represents the fitness value of the individual in the process of the mutation operation; $F_{max}$ corresponds to the maximum fitness value of the population individual, and $F_{min}$ represents the minimum fitness value of a individual. $F_{avg}$ represents the average fitness value of the population, and $P_{c1}$, $P_{c2}$, $P_{v1}$ and $P_{v2}$ are constant parameters within the interval.

Eq. (7) is used to calculate the larger fitness value of individuals. In this process, the individuals are randomly paired and the equation is as follows:

$$P_{Cm} = \begin{cases} \dfrac{P_{C1}}{1 + e^{-lC}}, F' \geq F'_{avg} \\ P_{C2}, F' < F'_{avg} \end{cases} \tag{7}$$

Then, $k_c$ can be expressed as follows in Eq. (8), which refers to adaptive crossover rate.

$$K_C = \frac{F' - F'_{avg}}{F_{max} - F'_{avg}} \tag{8}$$

In the Eqs. (7) and (8), $F'$ refers to the maximum fitness value of the parent individual based on the crossover operation; $F_{max}$ represents the maximum fitness of individuals; $F'_{avg}$ represents the average fitness of individuals whose fitness is higher than $F_{avg}$; $P_{c1}$ and $P_{c2}$ are constant parameters.

Eq. (9) is the mutation operation. $P_m$ represents the fitness of each individual in the population which can be calculated with below Eq. (9):

$$P_m = \begin{cases} \dfrac{P_{m1}}{1 + e^{-lm}} * P, F \geq F'_{avg} \\ P_{m2}, F < F'_{avg} \end{cases} \tag{9}$$

$K_m$ represents the adaptive mutation rate. The number of mutation points and the mutation rate can be calculated with Eqs. (9) and (10), respectively.

$$K_m = \frac{F - F'_{avg}}{F_{max} - F'_{avg}} \tag{10}$$

$$NUM = num * P \tag{11}$$

$F$ represents the fitness of individuals during the mutation operation process; $F_{max}$ refers to the maximum fitness of individuals during the same process; The average fitness value of the individuals can be expressed as $F'_{avg}$ whose fitness is higher than $F_{avg}$; $P_{m1}$ and $P_{m2}$ are two constant parameters. The largest number of mutation points can be calculated by Eq. (11) which is named *NUM*.

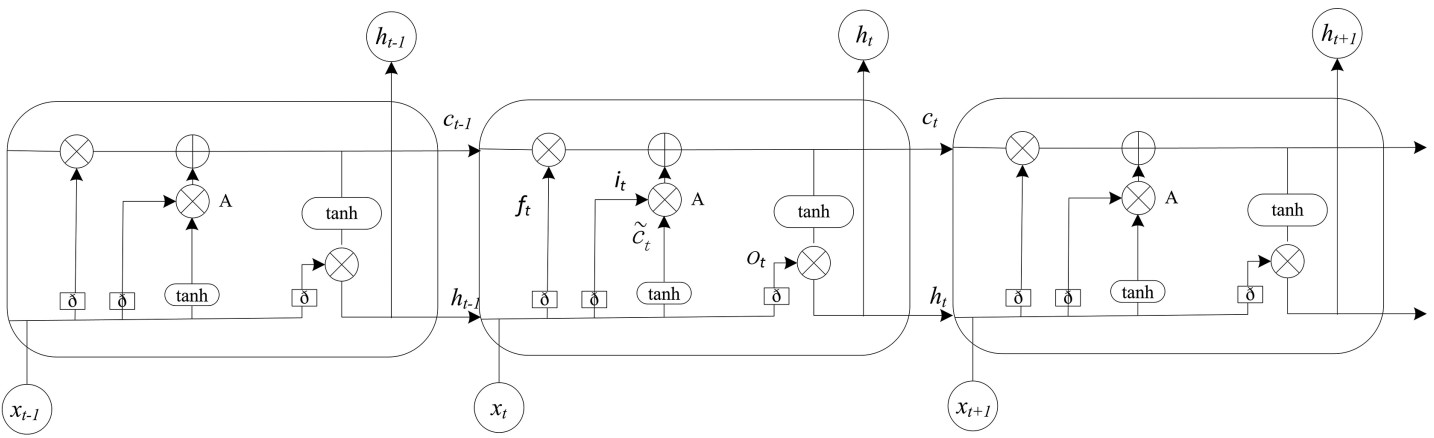

**Figure 2 The structure diagram of the LSTM.** The LSTM model consists of input gate, forgetting gate and output gate. Elements control the information flow in the hidden state through the gating unit.

The above operations improved the fitness value of the population, so that the fitness value can be raised to a higher level faster. The excellent genes will not be destroyed in the evolution, so it has a certain contribution to the improvement of the GA algorithm. This improved GA algorithm (IGA) will be applied to the optimization of parameters of the deep learning algorithm in this article.

## LONG-TERM AND SHORT-TERM MEMORY NEURAL NETWORK (LSTM)

The long-term and short-term memory neural network is also called gate recurrent neural network (gate RNN), also known as a recurrent neural network, including the network based on long-term and short-term memory (LSTM) and the network based on gate recurrent unit. The recurrent neural network can store the relationship between the input of the neuron at the current time and the output of the neuron at the previous time. In recent years, it has been used in nonlinear time series data prediction. Due to the limited depth of RNN in the time dimension, it is easy to produce gradient disappearance or gradient explosion when processing samples with many parameters and long-term time dependence. The LSTM well solves the problem of the long-term dependence of samples. Its network structure adds one input and one output based on the RNN. The gating unit can flexibly change the connection weight. It also improves the structure of repeated modules. Its prominent feature is to strengthen the control ability of information by using the threshold composed of a sigmoid neural network layer and point-by-point multiplication. As shown in Fig. 2, the forgetting threshold layer $f_t$ has the ability to selectively learn, to stop learning or to retain information from each unit. That determines whether it should be retained or discarded in the process of information transmission. The unit status to be updated is determined by the input threshold layer $i_t$. The output threshold layer $o_t$ will filter the output based on the unit state, and the update of each threshold layer. The calculation formula is shown in Eq. (12). Where $W_f$, $b_f$, $W_i$, $b_i$, $W_o$ and

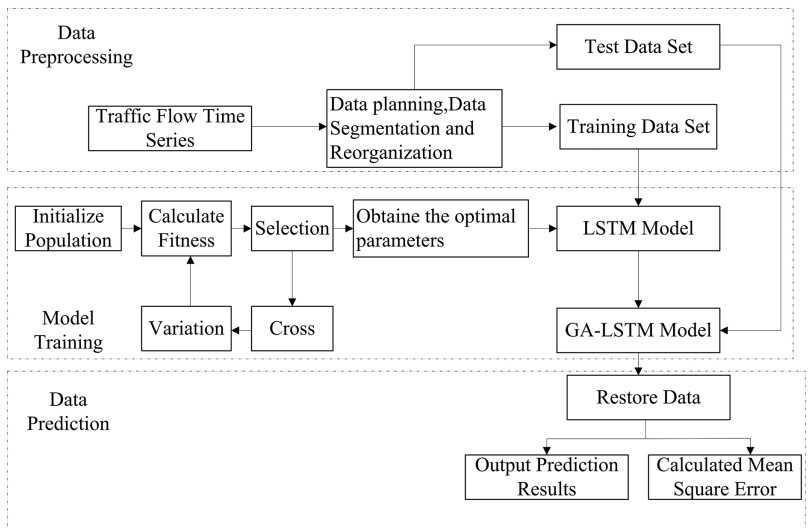

**Figure 3 Framework diagram of traffic flow forecast by the IGA-LSTM model.** The dotted line box represents the module, and the solid line box represents the data processing and operation steps. The data processing module needs to normalize and reorganize the traffic flow data set and divide it into training data set and test data set; In the model training part, the initialization parameters of the LSTM are optimized by improved the GA, so as to obtain the best parameter combination; The data prediction module is output. After restoring the data, the prediction results are output and compared with the test data set to calculate the Root Mean Square Error. 

$b_o$ are the weight and offset of each threshold layer respectively, σ represents sigmoid activation function.

$$f_t = \sigma(W_f \cdot [h_{t-1}, x_t] + b_f)$$
$$i_t = \sigma(W_i \cdot [h_{t-1}, x_t] + b_i) \quad\quad\quad\quad (12)$$
$$o_t = \sigma(W_o \cdot [h_{t-1}, x_t] + b_o)$$

The unique unit state and threshold layer of the LSTM extend the memory ability of the RNN model. In the training process of the neural network, the weights and offsets of each threshold layer are learned from the historical data set to identify and remember the characteristics of the historical state. In the real-time prediction stage, the prediction value of the time series can be obtained by calculating the input data based on the trained model (*Do, Taherifar & Vu, 2018*; *Polson & Sokolov, 2017*).

## THE IGA-LSTM MODEL

### Model establishment

The parameter selection of long-term and short-term memory neural network models has a great impact on the results. The improved genetic algorithm is introduced to select the parameters of the LSTM, and the IGA-LSTM traffic flow prediction model is established. The optimized parameters can reduce the impact of the initial parameter setting on the prediction results. The overall framework is shown in Fig. 3, including three modules: data processing, model training and data prediction. The improved genetic algorithm optimizes the parameters of the LSTM, including the number of hidden units, training times,

gradient threshold and learning rate. The weight of the model is adaptively adjusted through the LSTM iteration to form the IGA-LSTM model. After determining the optimal parameter combination, the traffic flow time series is input into the IGA-LSTM model, and the output value is the traffic flow prediction value. Through repeated experiments, the training data set is input into the model for operation, and the error between the prediction results and the test data set is compared and output.

## Data reprocessing

The measured traffic flow data is a nonlinear time series. The data is normalized after being imported by the IGA-LSTM model. The formula adopted is Formula (13).

$$d_i = \frac{f_{di} - \min\{f_{dj}\}}{\max\{f_{dj}\} - \min\{f_{dj}\}} \qquad 1 \leq j \leq n \tag{13}$$

where $f_{di}$ refers to the traffic flow data. $\max\{f_{di}\}$ and $\min\{f_{di}\}$ are the maximum and minimum values of the sample. The obtained data sequence is divided into training data set and test data set, which are expressed as $d_{tr} = \{d_1, d_2, \ldots, d_m\}$ and $d_{te} = \{d_{m+1}, d_{m+2}, \ldots, d_n\}$.

## Experiment platform

The computer configuration and software environment used in the experiment are as follows: the processor is Intel i5-6500, and the memory is 8.0GB; the system is Windows10 (64-bit); the programming language version is Python3.7; the IGA-LSTM model, the GA-BP model, the PSO-BP model and the LSTM model are implemented in the Keras library with Tensorflow as the backend.

## Model training and prediction results

The processed normalized data sequence is input into the IGA-LSTM model for training, set the population individuals to 50, the number of iterations to 200, the variation rate and crossover probability to be automatically adjusted, set the number of initially hidden unit layers of the LSTM model to 3-200, set the number of training epoch to 250 rounds, set the initial value of dropout layer to 0.25, take the time step of 1–10 and the step of 1. At each time step of the input sequence, the LSTM network learns to predict the value of the next time step. Specify an initial learning rate of 0.005 and reduce the learning rate by multiplying by a factor of 0.2 after 125 rounds of training. For each prediction, the previous prediction is used as the input of the function. The IGA-LSTM model can store better individuals and avoid the degradation of the population in the process of evolution. Through the analysis of the average fitness convergence curve of the model, we can see that as the number of iterations increases, the fitness value tends to converge steadily, and the error is in a downward trend, thus achieving the optimal solution of the search space. The model fitness convergence curve is shown in Fig. 4 and tends to be stable after 40 generations of iteration.

In this article, the measured data of three sections of the PEMs official data in California are selected for analysis. The data contains the flow data set of three road sections in the

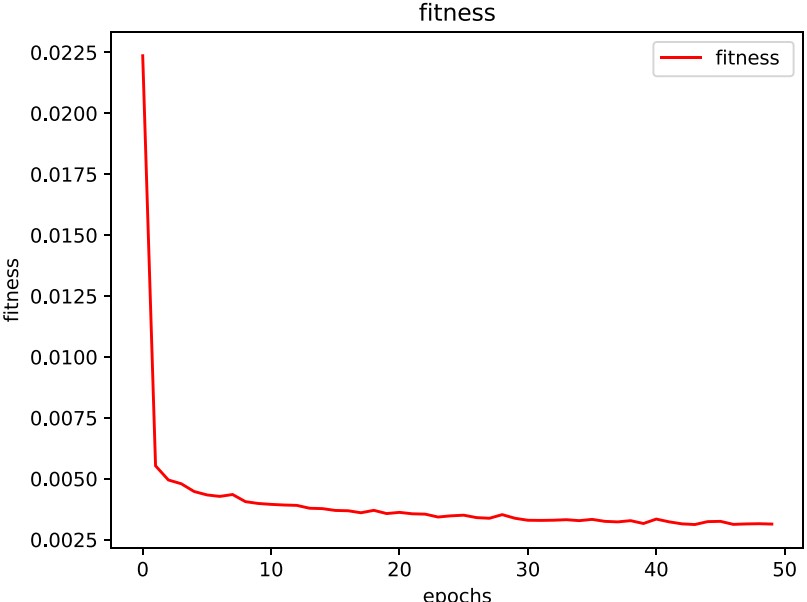

**Figure 4 Fitness convergence curve of the IGA-LSTM model.** The red curve represents the fitness convergence curve of the algorithm, and the average fitness is taken.

road network. Select the daily flow data set of the detector from June 1 to June 10, 2019, the sampling interval is 5 min, and the data set has a total of 3 * 7 * 288 data points. Record fields include: date, flow, number of lanes, occupancy and average speed. Zero data in the original data has been repaired, and the mean value of the data before and after the missing point is taken. The first 6 days of the selected data are the training data set, and the seventh day is the test data set. The Root Mean Square Error (RMSE), Standard Error (SE) and the Maximum Square Error (MMSE) are used as the evaluation index of prediction results.

To cooperate with the algorithm calculation, formula (6) is used to normalize the original data to reduce the data proportion after difference to the range of [−1,1]. The IGA-LSTM proposed in this article is used to train the data, and the seventh-day data is used for prediction. The prediction results are shown in Figs. 5A–5D.

Before and after using the IGA model to optimize and update the weight and threshold of the LSTM, the prediction results of normalized traffic flow data are compared. The error is reduced by 50%, and the prediction accuracy is greatly improved. After optimizing and fine adjusting the parameters of the LSTM in the process of model evolution, the final prediction results of the four algorithms are compared as shown in Fig. 6.

Compared with LSTM algorithm, the GA-BP algorithm and the PSO-BP algorithm, the IGA-LSTM algorithm proposed in this article can better fit the real data and achieve the prediction effect. Compared with the LSTM algorithm, the GA-BP algorithm and the PSO-BP algorithm, the IGA-LSTM algorithm has smaller output root mean square error, stronger model adaptability and better prediction accuracy in traffic flow data prediction.

In order to further verify the effectiveness of the algorithm, weekend data in the same period are used for simulation analysis. A total of 3 * 4 * 288 data sets are selected. 90% of

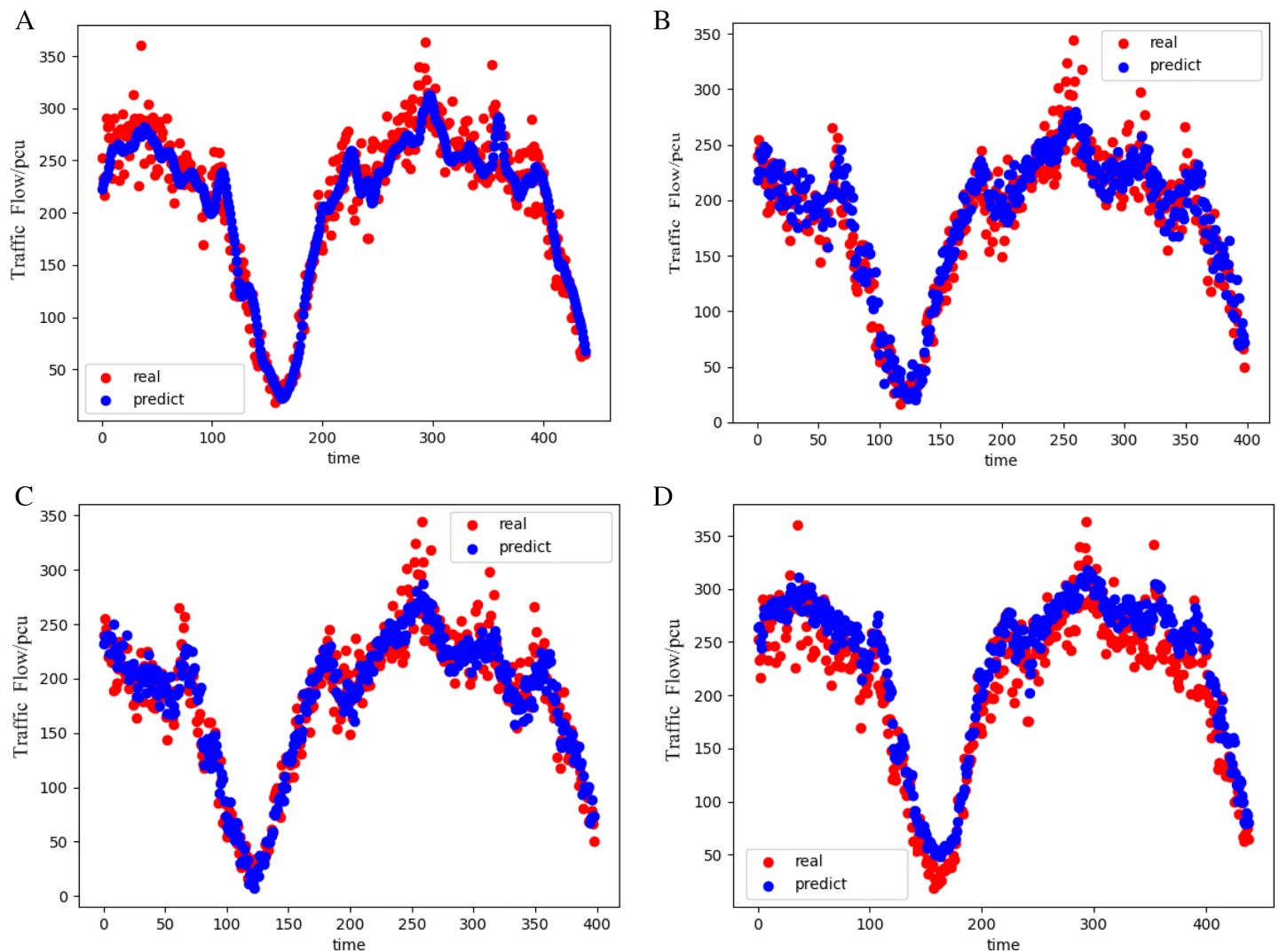

**Figure 5 Prediction results of the four models (weekday).** This is the prediction results of four models on weekdays. The red curve represents the real value on weekdays and the blue curve represents the predicted value on weekdays. (A) The results of the IGA-LSTM model on weekdays. (B) The results of the LSTM model on weekdays. (C) The results of the GA-BP model on weekdays. (D) The results of the PSO-BP model on weekdays.

the data are used for training the model and 10% of the data are used for testing. The prediction results of the four algorithm models are shown in Figs. 7A–7D. The final prediction results for weekend data of the four algorithms are compared as shown in Fig. 8.

## SIMULATION RESULTS AND ANALYSIS

For weekday data, because the periodicity of weekday traffic flow is obvious and the peak time of road traffic flow is relatively stable, the prediction results are better. By the simulation of four different models, the RMSE and Maximum Square Error (MMSE) of the data are shown in Table 1. The optimal model is the IGA-LSTM, followed by the LSTM, then the GA-BP and the PSO-BP. Because the LSTM has strong adaptability to time series

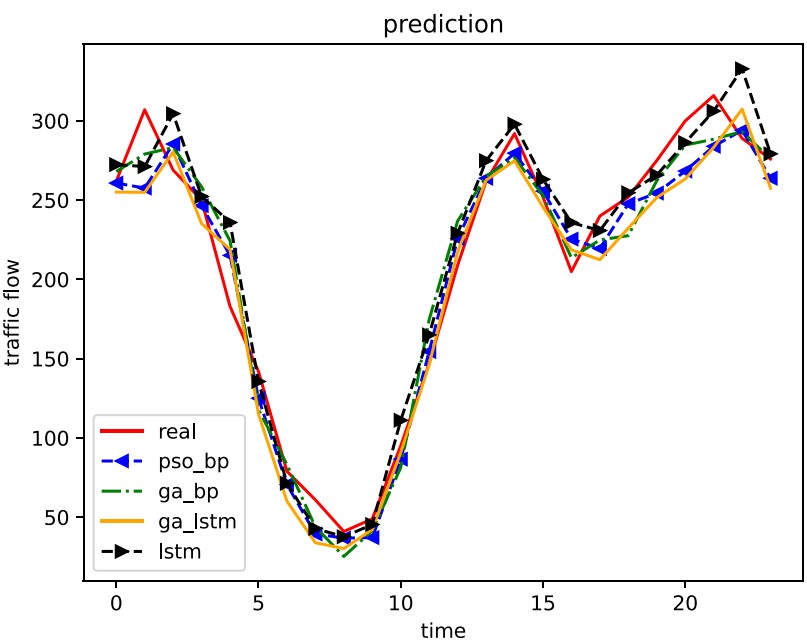

**Figure 6 Comparison of prediction results of four algorithms (weekday).** In the comparison curve of prediction results, red represents the test data set, green curve represents the prediction results of the GA-BP model, and blue represents the PSO-BP model prediction results, yellow represents the IGA-LSTM model prediction results, and black represents the LSTM model prediction results.

data, the performance of a deep learning algorithm is generally better than the traditional machine learning algorithm. Select the 1-h data from 8:00 to 9:00 in the morning peak of the test data on weekdays for analysis, intercept each 5-min data node respectively, and compare the root mean square error of the four algorithms, as shown in Fig. 9. It can be seen that the IGA-LSTM algorithm shows the smallest RMSE at the other 11 time nodes except the 7th-time and 9th-time nodes, and the advantages of the algorithm are very obvious. The RMSE of the four algorithms in time series is compared. The initial RMSE and maximum RMSE of the IGA-LSTM algorithm are the smallest, but the fluctuation is relatively large, and finally the average RMSE is the smallest; The RMSE characteristics of the GA-BP algorithm change smoothly, which shows that the error of the GA-BP algorithm changes little in the prediction process, and its average value is lower than that of the IGA-LSTM and the LSTM. The RMSE of the PSO-BP algorithm fluctuates greatly and the stability of the algorithm is poor.

For weekend data, the evaluation of the prediction results of the four models is shown in Table 2. From Table 2, it can be seen that the prediction accuracy of several algorithms in the prediction results of weekend data is lower than that of working days. From the analysis of the original data set, it can be seen that the periodicity of the weekend data set is less obvious than that of working days. The performance of the model in predicting such data is slightly poor. However, compared with similar data, the IGA-LSTM model still shows the best prediction accuracy and the lowest root means square error. The second is still the LSTM model, the GA-BP model and the PSO-BP model. Similarly, select the 1 h

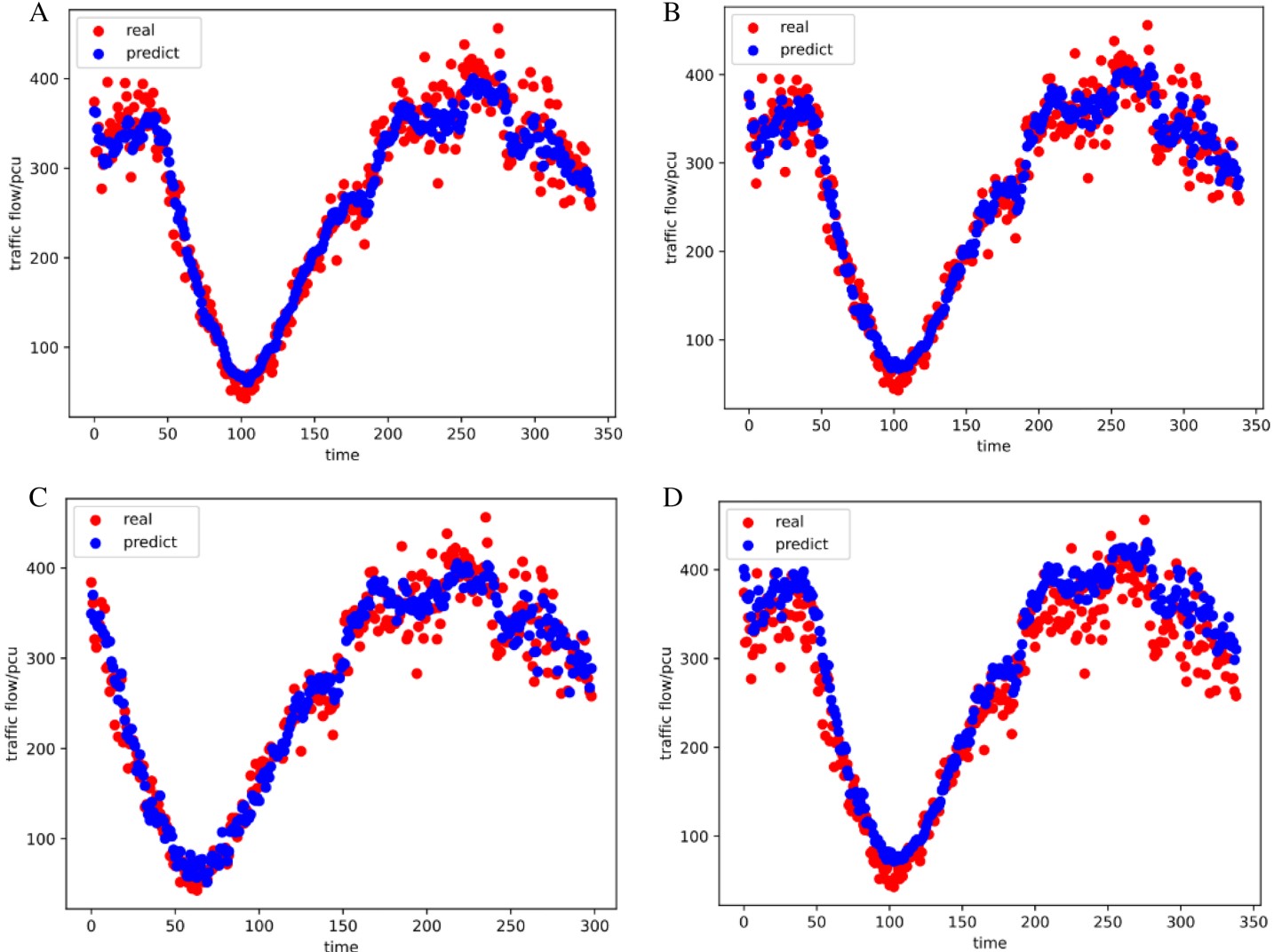

**Figure 7 Prediction results of the four models (weekend).** This is the prediction results of four models on weekends. The red curve represents the real value on weekends and the blue curve represents the predicted value on weekends. (A) The results of the IGA-LSTM model on weekends. (B) The results of the LSTM model on weekends. (C) The results of the GA-BP model on weekends. (D) The results of the PSO-BP model on weekends.

data of weekend between 8:00–9:00 AM for analysis, intercept each 5-min data node respectively, and compare the root mean square error of the four algorithms. As shown in Fig. 10, the IGA-LSTM algorithm shows the smallest RMSE feature at 8-time nodes, but it is still dominant. The root means square error characteristics of the PSO-BP algorithm are more optimized than the realization of working day data. The RMSE of the GA-BP algorithm is the largest at each time node.

The test results of integrating weekday and weekend data show that the IGA-LSTM algorithm model constructed in this article has better advantages in time dependence in all kinds of traffic flow data, high prediction accuracy and small output root mean square

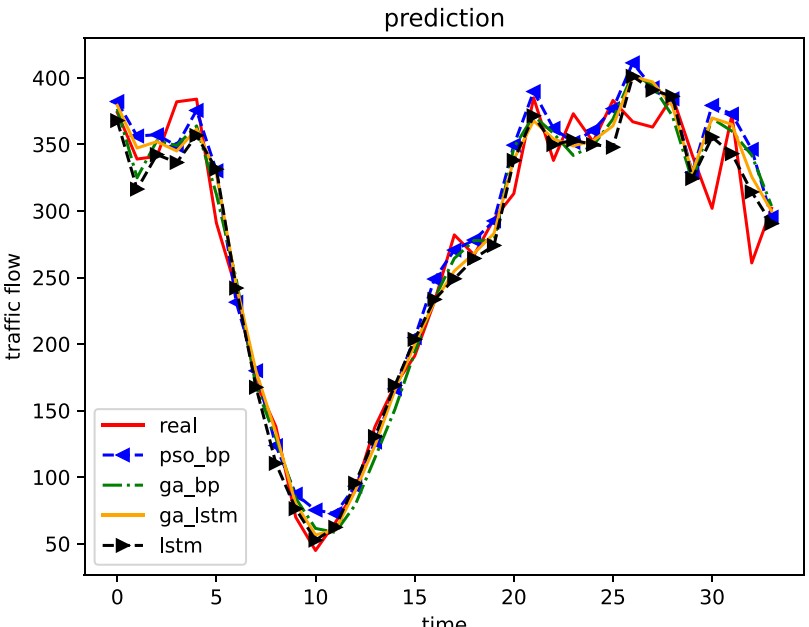

**Figure 8 Comparison of prediction results of four algorithms (weekend).** Select the representative 1-h time period between 8:00 and 9:00 in the morning and compare the root mean square error of the prediction results of the four algorithms every 5 min. In the figure, blue represents the GA-BP model, red represents the PSO-BP model, green represents the LSTM model and purple represents the IGA-LSTM model. It can be seen that the root mean square error of the IGA-LSTM model is the smallest on most nodes, Followed by the LSTM, then the GA-BP and the PSO-BP.

**Table 1 Evaluation of model prediction results (weekdays).** It can be seen that in the prediction of working day data, the Root Mean Square Error of the IGA-LSTM model is the smallest, followed by the LSTM model and the GA-BP model, and the Root Mean Square Error of the PSO-BP model is the largest.

| Model | RMSE | MMSE | SE |
|---|---|---|---|
| IGA-LSTM | 0.004466 | 0.00459 | 3.38076 |
| LSTM | 0.004538 | 0.1262 | 4.95769 |
| GA-BP | 0.00458 | 0.00773 | 4.13900 |
| PSO-BP | 0.00512 | 0.01407 | 3.94184 |

error, indicating that the algorithm has good prediction performance and good applicability (*Ali et al., 2017*).

All experiment results in this article are averaged through multiple runs. In order to show the comparison between the algorithms proposed in this article and other algorithms, the average Standard Error (SE) of the four algorithms are calculated more than 10 times. And the results are displayed as follows:

Figure 11 shows the calculation results of the weekday data samples. The standard error of all four algorithms are calculated. It can be seen that the IGA-LSTM algorithm still has considerable advantages compared with the other three algorithms. Among the selected samples, the standard error of the IGA-LSTM algorithm is the smallest in the proportion

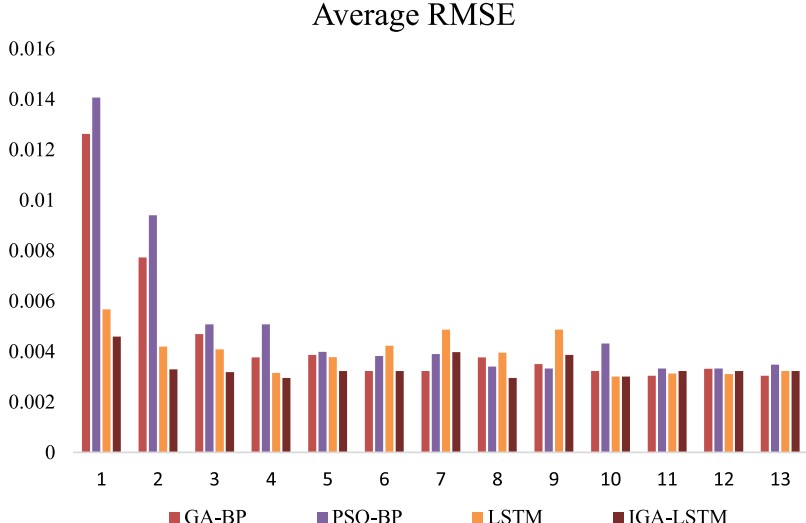

**Figure 9 Comparison of 1-h data time node RMSE (Weekday).** This is the Root Mean Square Error (RMSE) of the four algorithms of weekday. Red represents the RMSE of the GA-BP algorithm; Purplerepresents the RMSE of the PSO-BP algorithm; Yellow represents the RMSE of the LSTM algorithm; Brown represents the RMSE of the IGA-LSTM algorithm proposed in this article. The IGA-LSTM algorithm shows the smallest RMSE at the other 11 time nodes except the 7th and 9th-time nodes.

**Table 2 Evaluation of model prediction results (weekend).** It can be seen that in the weekend traffic flow data prediction, the Root Mean Square Error of the IGA-LSTM model is the smallest, followed by the PSO-BP model, then, the LSTM model and the GA-BP model. This result is different from the traffic flow prediction results on weekdays. The main reason is that the traffic flow peak data on weekdays has strong repeatability, and the repeatability of weekend data is not obvious.

| Model | RMSE | MMSE | SE |
| --- | --- | --- | --- |
| OGA-LSTM | 0.004675 | 0.00669 | 4.98384 |
| LSTM | 0.004988 | 0.00567 | 4.93000 |
| GA-BP | 0.005424 | 0.0109 | 6.33076 |
| PSO-BP | 0.00493 | 0.0133 | 5.04615 |

of 76.9%. Especially in the test of weekday data samples. Due to the strong periodic characteristics of traffic flow in weekdays, it can better reflect the superiority of the algorithm proposed in this article. Figure 12 shows the calculation results of the weekend data samples. Still the average standard error is multiple runs. But only about 53.8% of the selected samples have the smallest standard error by the IGA-LSTM algorithm. Therefore, in general, the algorithm proposed in this article has relatively higher prediction accuracy for weekday data, and relatively poor prediction accuracy for weekend data. Because the data in weekend has no obvious periodic characteristics.

# CONCLUSIONS AND RECOMMENDATION ON FUTURE WORKS

According to the spatiotemporal correlation of traffic flow series, a prediction method based on improved GA-LSTM model has been applied in this article. This method is based

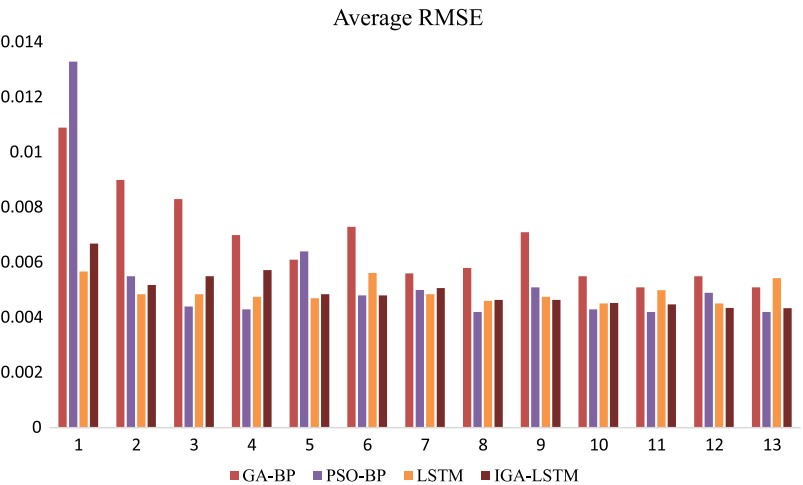

**Figure 10 Comparison of 1-h data time node RMSE (Weekend).** This is the Root Mean Square Error (RMSE) of the four algorithms of weekend. Red represents the RMSE of the GA-BP algorithm; Purplerepresents the RMSE of the PSO-BP algorithm; yellow represents the RMSE of the LSTM algorithm; brown represents the RMSE of the IGA-LSTM algorithm proposed in this article. The IGA-LSTM algorithm shows the smallest RMSE feature at 8-time nodes, but it is still dominant.

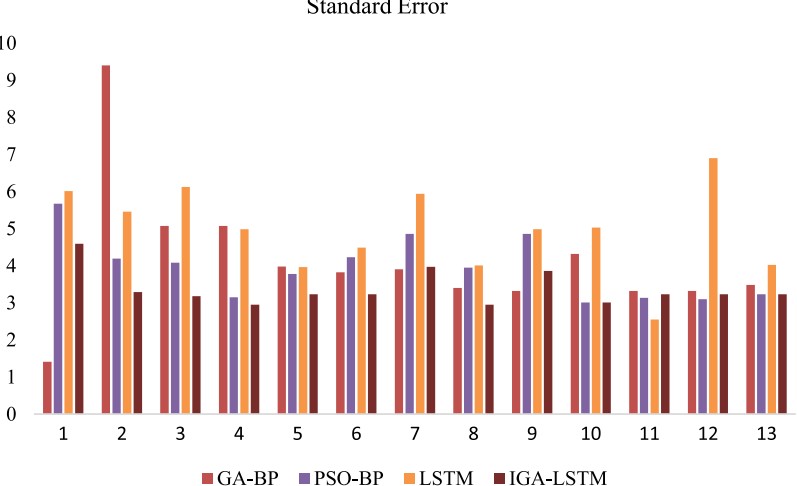

**Figure 11 Comparison of 1-h data time node SE (Weekday).** This is the Standard Error(SE) of the four algorithms on weekdays. Red represents the Standard Error of the GA-BP algorithm; purple represents the SE of the PSO-BP algorithm; yellow represents the SE of the LSTM algorithm; brown represents the SE of the IGA-LSTM algorithm proposed in this article. Among the selected samples, the Standard Error of the IGA-LSTM algorithm is the smallest in the proportion of 76.9%.

on long-term and short-term memory neural network, and adaptively adjusts the crossover rate and mutation rate of the GA through the improved genetic algorithm, so that the population can retain excellent genes in the process of evolution and will not be destroyed in evolution, to continuously improve the maximum fitness value of the population. The improved IGA model can optimize and adjust the number of hidden units, training times, gradient threshold, learning rate and other parameters of the LSTM

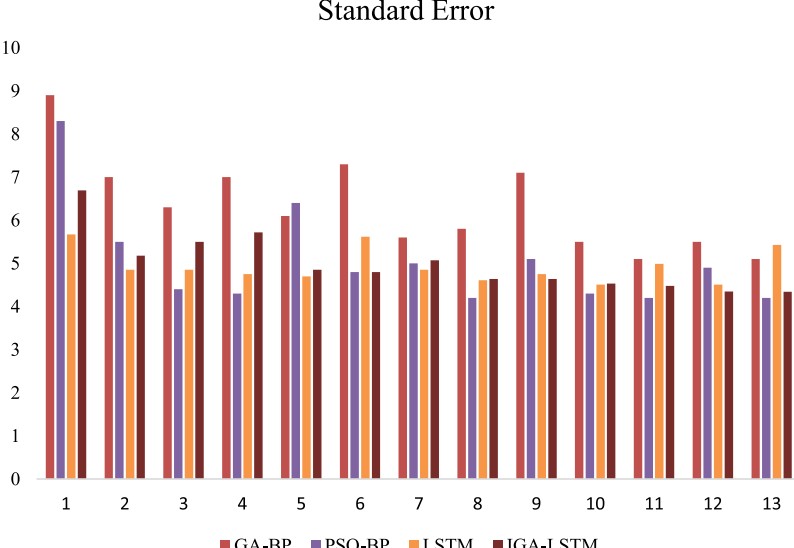

**Figure 12** **Comparison of 1-h data time node SE (Weekend).** This is the Standard Error (SE) of the four algorithms on weekends. Red represents the SE of the GA-BP algorithm; purple represents the SE of the PSO-BP algorithm; yellow represents the SE of the LSTM algorithm; brown represents the SE of the IGA-LSTM algorithm proposed in this article. About 53.8% of the selected samples have the smallest Standard Error by IGA-LSTM algorithm.     

model, so as to make the prediction accuracy of the LSTM model better and the mean square error smaller. The time complexity of the algorithm is increased $o(n^2)$, compared with pure deep learning algorithm. But the space complexity does not increase too much, and the prediction accuracy is improved by more than 10%. All abbreviated symbols and their meanings are shown in Table 3.

The experimental results show that the IGA algorithm can avoid population precocity and improve population fitness. By searching the optimal value of spatial parameters, the optimization efficiency is high. The training times have a great impact on the prediction results in the optimization process, followed by the number of hidden units. Too large hidden units will reduce the operation efficiency, and the accuracy value will not be improved, but too small number of hidden units will reduce the accuracy. The value of the dropout layer has little impact on the output error, and the adjustment of learning rate parameters has a great impact on the results. On the whole, compared with several commonly used machine learning algorithms, the IGA-LSTM model has higher operation efficiency, better accuracy and fast fitness convergence. Through empirical simulation in this article, it can be used for short-term traffic flow prediction of roads.

This algorithm mainly carries out targeted simulation experiments for short-term traffic flow data, but it can also be used for medium and long-term traffic flow prediction. Because of the strong feature extraction ability and big data processing ability of deep learning, it can further consider the traffic flow prediction of multi-state cross data at the upstream and downstream of the road section.

In the future, we will conduct data correlation analysis on the upstream and downstream traffic flow of the tested road. The impacts of the upstream and downstream

**Table 3 A notation table gives all abbreviated symbols and their meanings.**

| Abbreviation | Full name |
|---|---|
| ITS | Intelligent Transportation System |
| LSTM | Long-term and short-term memory neural network |
| IGA | Improved genetic algorithm |
| IGA-LSTM | Improved genetic algorithm optimize the long-term and short-term memory neural network |
| BP | Back propagation neural network |
| GA-BP | Genetic algorithm optimize the Back propagation neural network |
| PSO-BP | Particle swarm optimization algorithm optimize the Back propagation neural network |
| SVM | support vector machine |
| DBN | Deep belief network |
| CNN | Convolutional neural network |
| RNN | Cyclic neural network |
| GRU | Gated cyclic unit neural network |
| SAE | Stacked automatic encoder neural network |
| CNN-LSTM | Convolutional neural network and Long-term and short-term memory neural network |
| BiLSTM | Bi-directional Long and Short-Term Memory neural network |
| RMSE | Root mean square error |
| MMSE | Maximum square error |
| SE | Standard Error |

**Note:**
The full names of all abbreviation symbols in this article are shown in the table.

road will be fully considered on the traffic flow of the target road section. We will reconstruct the training data to obtain better data samples. That can improves prediction accuracy. At the same time, we also plan to consider the medium and long-term traffic flow forecast of the target road so as to provide support for the decision-making of the traffic management department.

### Funding
This work was supported by the Natural Science Foundation of China (No. 12004293). The funders had no role in study design, data collection and analysis, decision to publish, or preparation of the manuscript.

### Grant Disclosures
The following grant information was disclosed by the authors:
Natural Science Foundation of China: 12004293.

### Competing Interests
The authors declare that they have no competing interests.

## Author Contributions

- Junxi Zhang conceived and designed the experiments, performed the experiments, performed the computation work, prepared figures and/or tables, authored or reviewed drafts of the article, and approved the final draft.
- Shiru Qu analyzed the data, authored or reviewed drafts of the article, and approved the final draft.
- Zhiteng Zhang analyzed the data, performed the computation work, authored or reviewed drafts of the article, and approved the final draft.
- Shaokang Cheng performed the computation work, prepared figures and/or tables, authored or reviewed drafts of the article, and approved the final draft.

## Data Availability

The raw measurements are available in the Supplemental Files.

## Supplemental Information

Supplemental information for this article can be found online at http://dx.doi.org/10.7717/peerj-cs.1048#supplemental-information.

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
