# Peer review of "Improved genetic algorithm optimized LSTM model and its application in short-term traffic flow prediction"

_PeerJ Computer Science, doi:10.7717/peerj-cs.1048_

## Round 0.1 · original submission · Major Revisions

Based on the reviewers' comments, the authors are requested to make "major revisions" and resubmit the paper.

Reviewer 1 ·

Basic reporting

In this paper, author presented an improved genetic algorithm (OGA) to optimize long-term and short-term memory neural network (LSTM). However, there are some limitations that must be addressed as follows.
1. Some sentences in abstract are very lengthy (for example see sentence 1). These sentences should be changed to make the abstract more attractive for readers.
2. In Introduction section, it is difficult to understand the novelty of the presented research work. This section should be modified carefully. In addition, the main contribution should be presented in the form of bullets.
3. Different existing application of traffic events or traffic flow should be discussed. In addition, the most recent work about Deep learning-based traffic events or traffic flow should be discussed as follows (‘Traffic accident detection and condition analysis based on social networking data’, ‘Fuzzy Ontology and LSTM-Based Text Mining: A Transportation Network Monitoring System for Assisting Travel’, and ‘Transportation sentiment analysis using word embedding and ontology-based topic modeling’, and ‘Fuzzy ontology-based sentiment analysis of transportation and city feature reviews for safe traveling’.).
4. Equations should be discussed deeply.
5. LSTM is not properly discussed in Section 4. What about Bi-LSTM?
6. Captions of the Figures not self-explanatory. The caption of figures should be self-explanatory, and clearly explaining the figure. Extend the description of the mentioned figures to make them self-explanatory.
7. The whole manuscript should be thoroughly revised in order to improve its English.
8. More details should be included in future work.

Experimental design

no comment

Validity of the findings

no comment

Reviewer 2 ·

Basic reporting

The work is technically sound and has potential merit for publication. However, major revisions are needed to make it worth publishable.

Experimental design

The experimental section, in particular, the setup and parameters being used, needs further explanation.

Validity of the findings

The experiments seem inline, valid, and consistent with previous findings. However, the evaluation section is written well. However, it is not clear how the simulations were performed? Which optimization tool was used? Furthermore, the evaluation metrics should be briefly described.

Additional comments

I have the following comments and suggestions to further improve the quality of the work:

1) The abstract is not concise enough to sketch the entire theme, in particular, the results of the manuscript. Also, there were some serious grammar issues that should be corrected.

2) Furthermore, the introduction section needs considerable effort (concise and brief). The problem being investigated should be described clearly.

The introduction, e.g., should lead the way throughout the paper. In addition, the benefits coming from this paper should be made clearer in the introduction and throughout the paper.

- The entire introduction section is written in two paragraphs, while the first paragraph has been ended with a semi-colon (full stop needed)
- Furthermore, briefly describe the major contributions in bullet form, just before the organization paragraph.
- The final paragraph of the introduction section should be the organization flow.

3) I suggest adding a separate section that illustrates the related work. Various sections in the paper should be moved to this section. Moreover, a summary of the related work should be sketched into a table with respect to their characteristics. The authors should put their proposal into this table for easy comparison.

This will make it clearer to readers and they will be able to see what was missing in the literature; and how this is addressed in this paper.

4) Moreover, It would be better to add all notations in a Table for easy understanding.

5) The organization of the paper should be improved. For example, Sec. 5 has the proposed technique as well as results. I suggest putting them into two separate sections.

6) Does the results shown in various tables and figures refer to multiple runs (average)? In the latter case, I will suggest adding standard deviation bars. The reason behind this is to ensure that whether the results overlap with the closest rivals or not.

7) How about the time and space complexity of the proposed algorithms?

8) The evaluation section is written well. However, it is not clear how the simulations were performed? Which optimization tool was used? The experimental section, in particular, the setup and parameters being used, needs further explanation. Furthermore, the evaluation metrics should be briefly described.

9) I suggest adding pseudo-code for the proposed algorithm. Various procedures in the algorithm needs explanation. I suggest step-by-step details of the algorithm.

10) English can be improved. Proofreading should ensure the appropriate use of grammar, tenses, and punctuations. Longer sentences should be converted into smaller ones. Many words are missing their articles (the, a, an). Many punctuation characters are missing, and some are redundant.

---

## Round 0.2 · Minor Revisions

Based on the reviewers' comments, Kindly make "minor revisions" and resubmit the paper.

Reviewer 1 ·

Basic reporting

The authors have addressed my comments properly. Therefore, this paper can be accepted in its present form.

Experimental design

no comment

Validity of the findings

no comment

Reviewer 2 ·

Basic reporting

The revised version is much better than the original submission. However, structure of the manuscript should be improved.

Experimental design

The authors have improved their experimental discussion in the revision stage.

Validity of the findings

The manuscript is well written, and the conclusions are supported by the experiments.

Additional comments

Please address the following comments:

1) I suggest adding a paragraph that describes the organization of the paper in the introduction section. Usually, this should appear just after the contribution list.

2) Furthermore, some sections are not numbered, while others are labeled incorrectly. I suggest that authors be consistent all over the paper.

3) The paper desires proofreading and substantial struggles to correct grammar, linguistic, and punctuation errors. I cannot list them all, but the majority are punctuation mistakes. Some sentences do not make sense at all and should be rephrased.

---

## Round 0.3 · accepted · Accept

Based on the reviewers' comments and recommendations, the paper is accepted.

Reviewer 2 ·

Basic reporting

Well improved.

Experimental design

Well written.

Validity of the findings

The revised version has potential merit for publication.

Additional comments

N/A